# Psychological Status of Men Who Have Sex with Men during COVID-19: An Online Cross-Sectional Study in Western China

**DOI:** 10.3390/ijerph20021333

**Published:** 2023-01-11

**Authors:** Bing Lin, Jiaxiu Liu, Wei He, Haiying Pan, Xiaoni Zhong

**Affiliations:** 1School of Public Health, Chongqing Medical University, Chongqing 400016, China; 2Research Center for Medicine and Social Development, Chongqing 400010, China; 3School of Medical Informatics, Chongqing Medical University, Chongqing 400016, China

**Keywords:** COVID-19, psychological status, MSM, PQEEPH, China

## Abstract

Background: The psychological status of men who have sex with men (MSM) as a vulnerable population during COVID-19 is worthy of attention. However, studies of Chinese MSM are limited. The aim of this study was to investigate the psychological status and influencing factors of MSM population and to provide a scientific basis for this group to actively respond to public health emergencies. Methods: From June to September 2020, we conducted an online survey. MSMs were recruited through collaboration with non-governmental organizations (NGOs) and peer recommendations. The variables we collected included four aspects: demographic and HIV-related characteristics; COVID-19-related knowledge, attitudes, and behaviors; COVID-19-related risk perception; epidemic exposure. The psychological status was assessed by the Psychological Questionnaire for Emergent Events of Public Health (PQEEPH), which defined the psychological status as five primary disorders: depression, neurasthenia, fear, anxiety, and hypochondria. Multivariate logistic regression was used to explore the influences of COVID-19-related factors on the psychological status. Results: We surveyed 412 MSM online during COVID-19. The five psychological status of high-risk states accounted for 16.99% (depression), 14.08% (neurasthenia), 16.75% (fear), 16.50% (anxiety), and 18.20% (hypochondria), respectively. Not being well-informed about the causes of COVID-19 (*p* = 0.020) and having experienced epidemic exposure (*p* = 0.006) were able to promote the occurrence of depression. Lack of knowledge of the curative effect of COVID-19 has a higher risk of occurrence neurasthenia (*p* < 0.001). Being afraid of the novel coronavirus (*p* < 0.001) promoted fear. The perceived prevalence of the epidemic (*p* = 0.003), being more susceptible to COVID-19 (*p* < 0.001), and not being well-informed about the causes of COVID-19 (*p* = 0.005) had a positive effect on anxiety. Considering that the epidemic was not effectively controlled (*p* = 0.017), being more susceptible to COVID-19 (*p* < 0.001) was a contributing factor to the hypochondria. Conclusions: The incidence of psychological disorders in Chinese MSM was higher than that in other groups during COVID-19. Factors associated with COVID-19 may cause a range of mental health problems in this population. Greater attention should be paid to the mental health status of special populations during the epidemic, and effective preventive education and intervention measures should be taken.

## 1. Introduction

Men who have sex with men (MSM) have become the population with the highest risk of human immunodeficiency virus (HIV) infection and the fastest growing HIV epidemic in China [1,2]. Although same-sex sexual behavior is not illegal in China, MSM still face serious negative social and cultural consequences [3]. Due to the influence of traditional Chinese culture, MSM have conflicts in their self-identity, resulting in low self-acceptance, negative sexual attitudes, and great psychosocial pressure [4]. Compared to the general population, MSM are a more sensitive and vulnerable group with a higher probability of mood disorders, anxiety, and depression [5,6]. Mills [7] mentioned in his study that MSM are 2.7 times higher than the average male when the same tool was used to compare distressed and depressed men. The same findings were also seen in previous study, where MSM had a ratio of 2.9 for all mood disorders compared to other men [8]. Meanwhile, meta-analysis showed that the prevalence of depression in the Chinese MSM population was 43.9% [9], significantly higher than the 3% in men [10]. More importantly, poor mental health may also lead to an increased risk of HIV infection in MSM [11]. For example, previous studies have shown that lower levels of mental health in MSM are significantly and positively associated with unprotected anal intercourse [12], and severe cases can also lead to suicidal ideation [13,14]. Therefore, it is obvious that strengthening MSM’s mental health is important for enhancing their quality of life and social adjustment.

Since the first outbreak of COVID-19 in early 2020, a variety of strict public health measures have been implemented across China, such as regular lockdowns, requirements for distance, and reduced public services. A prolonged lockdown and a large number of new cases will not only affect the social economy [15,16], but also has the potential to disrupt people’s normal lives and lead to work and social dysfunction, for example, by causing a range of mental health problems [17,18]. Additionally, previous studies have shown that the rapid spread and high mortality of the severe acute respiratory syndrome corona virus 2 (SARS-CoV-2) may also exacerbate the risk of mental health problems and worsen existing psychiatric symptoms, further impairing people’s daily functioning and cognitive abilities [19].

MSM are likely to be socially disadvantaged, and we believe that the COVID-19 epidemic also poses a significant mental health challenge for them. Recent literature suggests that COVID-19 may disproportionately affect sexual minority males, which includes MSM [20]. In fact, the daily lives of MSM, HIV-related health services, HIV pre-exposure prophylaxis (PrEP), and sexuality are all significantly impacted by sudden COVID-19 outbreaks. A survey of MSM during COVID-19 revealed that this epidemic not only cost them financially, but also interrupted their HIV prevention (PrEP), testing, health care, and treatment services [21]. According to the findings during the Australian lockdown, 1 in 6 MSM stopped using PrEP each day [22]. Additionally, an online study that was conducted in more than 100 countries revealed that social distance had a negative effect on the quality of some MSM’s sexual lives and that the number of MSM who had sex with a stable partner during COVID-19 decreased significantly by a factor of six [23]. Because of the prolonged lockdown, MSM may increase their use of dating and social networking applications after the lockdown is lifted or in defiance of the lockdown policy, thereby engaging in sexual activity that increases the risk of COVID-19 infection [24]. The changes in various aspects of MSM’s lives due to the COVID-19 epidemic may cause or even exacerbate a range of psychological problems in MSM and have a negative impact on their mental health. Improving the mental health of MSM may not only improve their ability to cope during public health emergencies but may also go some way to preventing the spread of HIV and other sexually transmitted diseases.

On the other hand, related theories are able to explain the psychological status in MSM population during COVID-19. According to the minority stress theory, stigma, prejudice, and discrimination against MSM population can produce stress that leads to negative health outcomes [25,26]. Minority stress processes include distal stressors, such as violence and discrimination related to personal identity, and proximal stressors, such as self-perception and evaluation [25]. For some MSM individuals, both distal and proximal stressors may be amplified to avoid the spread of COVID-19 in isolated situations such as government restrictions [27]. These amplified stressors, along with other uncertainties and fears associated with a pandemic, may increase mental health problems in the MSM population.

In conclusion, the COVID-19 outbreak, a serious public health emergency, not only resulted in significant losses in terms of human life and property, but also had a psychological impact on special populations. The physical damage may be recovered in a short period of time, but the negative psychological impact on a person may be lasting and persistent. The psychological status of the MSM population as a vulnerable group during this period deserves attention and can provide a theoretical basis for the subsequent psychological guidance work. Several countries, including the United States [28], Brazil [29], and Mexico [30], have conducted studies related to the mental health of MSM during COVID-19. Studies have been conducted in China to assess the mental health of different populations, including elderly [19,31], healthcare staff [32], and university students [33]. However, studies on the psychological status of Chinese MSM during COVID-19 are limited. Since MSM are at high-risk for both HIV and psychological disorders, it is necessary to assess their psychological status during COVID-19, considering the particularity of this population.

Our study conducted an online cross-sectional survey to assess the psychological status (including five psychological dimensions: depression, neurasthenia, fear, anxiety, and hypochondria) of MSM in Western China during COVID-19 by the Psychological Questionnaire for Emergent Events of Public Health (PQEEPH) and explored the influencing factors. An essential and significant component of the government’s mission is the effective management of psychological concerns resulting from public health emergencies. The simplest and most direct method for determining MSM’s psychological status following an event is to administer a psychological questionnaire to them. These questionnaires can screen high-risk populations for emotional bias or impairment and prompt timely psychological intervention to prevent the serious and further development of mental health issues. From the perspective of the MSM population’s awareness of public health emergencies and their knowledge reserve ability, it is useful to outline the psychological influencing factors of MSM during COVID-19, understand and predict possible psychological behavioral characteristics, and collect information on their psychological status so as to implement behavior change intervention strategies in a targeted manner.

## 2. Methods

### Study Design, Population, and Sampling

Our study was an online, cross-sectional survey conducted from June to September 2020 in Western China (Chongqing, Sichuan, Xinjiang). The participants of the study were from the National Key Project for Infectious Diseases of the Ministry of Science and Technology of China in the 13th Five-Year Plan. Subject recruitment and sampling are challenging in social studies and research involving gender and sexual minorities. Subjects often tend to prefer not to reveal themselves, and they also choose to hide or keep their true gender identity as a secret, or even attempt to avoid investigation [34]. Therefore, in our study, MSMs were recruited through collaboration with non-governmental organizations (NGOs) and peer recommendations. We promoted it in WeChat groups and QQ groups (the social media app in China) and worked with local NGOs to provide them with information on HIV prevention, counseling, and testing on the Internet. At the same time, we informed NGO managers of the purpose, process, potential benefits, risks, and other specific information of this study to gain their support.

The specific inclusion criteria were being male (physiological sex was male), being 18 years old or older, and self-reported having had sex with a male sexual partner (including casual or regular partners) in the past 6 months. The electronic questionnaire was distributed one-to-one to respondents via WeChat and QQ groups and returned to the researcher upon completion. If a respondent encounters a problem that was not clear, the investigator will explain it one-on-one. To avoid multiple submissions, each IP address could only fill out the questionnaire once. The questionnaires will be quality-controlled by the researchers. The questionnaires will be considered invalid if the total time taken to complete the questionnaire was less than five minutes or more than one hour (the time taken to answer each questionnaire can be monitored in the backend of the online questionnaire).

## 3. Measures

### 3.1. Demographic and HIV-Related Characteristics

Many previous studies have reported that socio-demographic characteristics, sexual orientation, and other factors are associated with anxiety and depression in MSM [35,36]. Meanwhile, previous studies have focused on MSM’s sexual identities (sexual role) [37]. Therefore, our study collected demographic and HIV-related characteristics and explored the potential impact of these factors on the psychological status of the MSM population. The demographic and HIV-related characteristics of the participants mainly included age, household location, ethnicity, educational level, employment status, marital status, monthly personal income, sexual role, number of male sexual partners in the last month, and self-reported HIV infection status. Among them, monthly personal income was calculated in RMB, the legal tender of China, and the unit is Yuan. The sexual role is the way of having penetrative sex with a male partner, such as the inserter (like “Top”) and the receiver (like “Bottom”).

### 3.2. COVID-19-Related Knowledge, Attitudes, and Behaviors

Knowledge, attitudes, and behaviors related to COVID-19 were measured primarily through a series of questions. Participants were asked 6 questions to measure their knowledge, including: “I am well-informed about the causes of COVID-19, transmission routes of COVID-19, infectiousness of COVID-19, effectiveness of preventive measures for COVID-19, effectiveness of cure of COVID-19, and reinfection of COVID-19 after cure”. Twelve questions were used to measure participants’ attitudes, including: “I think COVID-19 is very contagious”, “I think COVID-19 is very prevalent where I live”, “I am afraid of COVID-19”, “I think COVID-19 is very close to me” and so on. All questions were scored from 1 to 5, from completely disagree (1) to completely agree (5). Behavior associated with COVID-19 was measured by asking participants if they had been tested for the novel coronavirus.

### 3.3. COVID-19-Related Risk Perception

According to the SARS risk perception questionnaire of Burg [38], we measured the COVID-19-related perceived risk in the MSM population in four dimensions: controllability, possibility, susceptibility, and severity. Controllability consisted of three questions: “I can take protective measures such as home isolation, wearing masks and washing hands regularly; I can ensure that I am not infected with COVID-19; and I can control the loss caused by COVID-19”. Possibility included a question on the “I’m very likely to get infected with COVID-19”. Susceptibility included the question “I am more likely than others to get COVID-19”. Severity included the question “I think COVID-19 is very serious”. All questions were assigned a score from 1 to 5, ranging from completely disagree (1) to completely agree (5).

### 3.4. Epidemic Exposure

The epidemic exposure scale was developed on the basis of the exposure scale in the previous study in conjunction with COVID-19 outbreak [39,40]. Eight main questions were included: “Have been quarantined for confirmed or suspected patients with COVID-19”, “Do you have symptoms of suspected infection, such as fever, nasal congestion, runny nose, sore throat, and so on”, “Have any relatives or friends were quarantined because they were suspected to be patients with COVID-19”, “whether any relatives or friends were confirmed to be patients with COVID-19”, “whether any neighbors were quarantined because they were suspected to be patients with COVID-19”, “whether any neighbors have been diagnosed with COVID-19”, “whether there have been confirmed or suspected infected persons in the neighborhood where you live or in the surrounding area”, “the necessities has been a shortage due to the COVID-19 outbreak”. Participants who answered “yes” were given a score of 1, and those who answered “no” were given a score of 0. The scores of the 8 items were added together to obtain a total score. A score of 0 means no epidemic exposure, and a score greater than or equal to 1 means epidemic exposure.

### 3.5. The Psychological Questionnaire for Emergent Events of Public Health (PQEEPH)

The PQEEPH was used to measure the psychological status of the MSM population during the COVID-19 outbreak in our study (Table 1). The scale has been widely used to investigate the mental health status of individuals in public health emergencies [41,42,43]. The PQEEPH consists of 25 items assessing 5 psychological dimensions: depression (6 items), neurasthenia (5 items), fear (6 items), anxiety (6 items), and hypochondria (2 items). All items are scored on a 4-point scale ranging from 1 (occasionally) to 4 (always). The higher the total score for each dimension, the more severe the psychological symptom represented by that dimension. For each dimension of mental status, a score above the mean plus one unit of standard deviation was considered the cut-off point for a higher risk of mental health problems [42,43], which is considered an emotional deviation and is more likely to reach an unhealth mental status. Cronbach’s α = 0.961 for PQEEPH in our study and 0.934, 0.905, 0.774, 0.910, and 0.708 for the five dimensions, respectively. The content validity index (CVI) of PQEEPH was 0.95, and the CVI of the items ranged from 0.83 to 1.00 [43].

### 3.6. Statistical Analysis

The demographic and HIV-related characteristics of the participants were described. The mean and standard deviation of each dimension of mental status of the MSM population were calculated by PQEEPH. The mean value of each dimension of mental status plus one unit of standard deviation indicates the high-risk value of mental health, with values greater than the high-risk value being recorded as 1 and values less than the high-risk value being recorded as 0 and grouped as such. The variability of independent variables across groups in each psychological dimension was compared, and variables with *p* ≤ 0.1 were included in the subsequent models (Chi-Square Test in case of categorical variables and Trend Chi-Square Test in case of ordered categorical variables). A multivariate logistic regression model was developed to explore the effects of COVID-19-related factors on each psychological dimension of the MSM population, using high-risk values for each psychological dimension were achieved as dependent variables, COVID-19-related knowledge, attitudes and behaviors, risk perception, and epidemic exposure experience as independent variables, and correcting for demographic and HIV-related characteristics. Stepwise regression adopted backward selection method, with slentry = 0.05 as the criterion for variable inclusion and slstay = 0.05 as the criterion for variable exclusion. The results were expressed as crude odds ratios (cOR) and adjusted odds ratios (AOR) with 95% confidence intervals (CI). The *p*-value < 0.05 was considered statistically significant. All statistical analyses were done using SAS V.9.4 software.

## 4. Results

A total of 456 questionnaires were collected in this online survey, 44 of which were excluded due to short or long response time, and 412 questionnaires were included in the final analysis, with a valid return rate of 90.35%. Of the 412 MSM surveyed, 60.19% were aged 18–35 years old, 34.47% were aged 35–50 years old, and 5.34% were older than 50 years old. Overall, 94.17% of the MSM were from urban areas, and 90.53% were Han ethnicity. Other demographic and HIV-related characteristics are shown in Table 2. 

The scores of each psychological dimension of MSM population were calculated according to PQEEPH (Table 3). Finally, we obtained that the means and standard deviations of depression, neurasthenia, fear, anxiety, and hypochondria were 9.91 ± 4.33, 8.17 ± 3.50, 11.27 ± 3.64, 8.44 ± 3.64, and 2.73 ± 1.14, respectively. Using the mean value plus one unit of standard deviation to represent the high-risk status for that psychological dimension, 16.99% (n = 70), 14.08% (n = 58), 16.75% (n = 69), 16.50% (n = 68), and 18.20% (n = 75) of the MSM population were at high-risk state for each psychological dimension, respectively. The variability of the factors associated with COVID-19 in each psychological dimension grouping was compared separately, and variables with *p* ≤ 0.10 were included in the multivariate logistic regression analysis (Table 4). The complete univariate analysis form, including different groups of all variables, is attached in the Appendix A.

According to the results of multivariate logistic regression analysis, after correcting for the demographic and HIV-related characteristics of the MSM population, the COVID-19-related factors were significantly associated with each dimension of mental status (Table 5). Of these, MSM who were not well-informed about the causes of COVID-19 (AOR = 2.69, 95% CI: 1.06–6.80, *p* = 0.020) and who had experienced epidemic exposure (AOR = 2.26, 95% CI: 1.27–4.02, *p* = 0.006) were more likely to be depressed. Lack of knowledge of the curative effect of COVID-19 has a higher risk of occurrence neurasthenia (AOR = 9.59, 95% CI:2.29–32.54, *p* < 0.001). Being very afraid of the novel coronavirus was significantly associated with the development of fear (AOR = 4.71, 95% CI: 1.27–17.56, *p* < 0.001). A perceived very high prevalence of COVID-19 epidemic in the place of residence (AOR = 2.79, 95% CI: 1.12–6.92, *p* = 0.003), being more susceptible to COVID-19 than others (AOR = 8.54, 95% CI: 2.60–28.06, *p* < 0.001), and not being well-informed about the causes of COVID-19 (AOR = 4.02, 95% CI: 1.32–12.24, *p* = 0.005) were contributing factors to anxiety. MSM who believed that the epidemic was not effectively controlled (AOR = 28.47, 95% CI: 2.59–312.67, *p* = 0.017) and more susceptible to COVID-19 than others (AOR = 4.81, 95% CI: 1.70–13.58, *p* < 0.001) were more likely to be hypochondriacs.

## 5. Discussion

Our study measured the psychological status of MSM in Western China during COVID-19 and identified mild mental health problems in the MSM population (the proportion of high-risk status for the five psychological dimensions ranged from 14.08% to 18.20%) and suggested the possibility that factors associated with COVID-19 influenced the mental health of this population. In the meantime, the prevalence of psychological status disorders in the MSM population in Western China was significantly higher than in the general adult population in China (from 7.9% to 10.40%) [43], and in the elderly (from 10.00% to 14.90%) [42], compared to previous studies that used the same measurement instrument (PQEEPH) for the psychological status survey of the population during COVID-19. It is evident that the potential mental health problems of MSM population as a vulnerable group during the COVID-19 cannot be ignored. Meanwhile, unlike other studies, for MSM as a sexual minority population, we corrected not only for demographic characteristics but also for HIV-related characteristics. In particular, two variables, sexual role, and number of male sexual partners, were included in multiple regression models. The comparison of cOR and AOR in the regression model also allows us to see that neglecting to correct for these factors will lead to misestimation of the psychological status of the MSM population. Some characteristic factors (especially sex-related variables) cannot be ignored in the study of the psychological status of this particular group of MSM population.

The logistic regression analyses showed that risk perception of COVID-19 promotes anxiety and hypochondria in MSM population, highlighting the significant impact of susceptibility factors in risk perception on mental health. Studies conducted during COVID-19 have reported that risk perception was a significant predictor of public mental health [44,45]. Higher levels of risk perception are a risk factor for mental illness [46]. Several social media monitoring studies documented public online data during COVID-19 outbreak and revealed a correlation between risk perception and negative emotions [47,48]. These previous studies support our findings regarding the MSM population. Therefore, in response to our findings, the government and management should take necessary interventions to reduce the risk perception of MSM population and improve social support, thus improving their mental health.

In the meantime, our findings also reported that not being well-informed about the causes of COVID-19 promotes anxiety and depression in MSM. This may be due to the lack of knowledge related to COVID-19 among the MSM population. For instance, an empirical study had pointed out that knowledge on COVID-19 positively influence depression [49]. Other studies have also demonstrated that lack of awareness of the virus can lead to depression and have a negative impact on mental health [50]. A high level of knowledge can maintain situational awareness, which not only helps to avoid infection but also improves mental health. Hence, management should enhance health promotion among the MSM population, which will help them understand the knowledge related to COVID-19 so as to reduce mental illness. 

Our findings suggested that epidemic exposure, a significant stressor, may adversely affect the mental health of MSM population. Previous studies of SARS have shown that increased exposure can affect mental health, and people who are isolated in high-risk environments and have family or friends with SARS have significantly higher levels of post-traumatic stress symptoms than those who do not have these experiences [39]. This is consistent with our findings. We believe that the theory describing how risky events affect mental health can be explained by the “ripple effect”. That is, the closer one is to the center of the event, the higher one’s risk perception and negative emotions are in the face of a risk event [51]. Previous studies have shown that mental health services are critical to mitigate the mental health impacts associated with epidemics [18]. This suggests that management should pay attention to the mental health of MSM population during COVID-19 epidemic, especially MSM who have experienced epidemic exposure, and carry out timely psychological services and interventions to help them improve their ability to cope with major stressors and improve their mental health.

In summary, in order to improve the psychological status of the MSM population during COVID-19, the government and management should take the necessary interventions. However, MSM population, as a vulnerable group, are not always willing to be identified and sought out as a high-risk group for mental illness. Therefore, interventions for this special population should be differentiated from those for the general population. On the one hand, management should cooperate with local NGOs and use these organizations as intermediary agents to educate the MSM population and add psychological counseling as a service component to the regular HIV health services. Strengthening support for these intermediary organizations (including policy support and financial support) and ensuring the normal operation during COVID-19 will help increase the MSM population’s sense of security and trust in the organizations, and thus provide them with adequate services and assistance. On the other hand, targeting social apps frequently used by MSM population (for example, Blue [52]) and distributing information on HIV prevention, counseling, and testing, COVID-19-related health promotion, and psychological counseling through online resources can maximize the scope of outreach and education among MSM population. Making full use of online resources, online research and studies can be used to develop targeted and reasonable psychological interventions, such as psychological status assessment, psychological guidance, and individual behavioral interventions, so that MSM are fully aware of the importance of good mental health, strengthening their awareness of psychological adjustment and improving their mental health problems.

As a cross-sectional survey of the mental health status of the MSM population conducted during the COVID-19 epidemic, our study has some shortcomings as follows. Firstly, our study used an online survey, and the quality of the questionnaire may not be as good as that of a face-to-face survey. However, conducting a face-to-face survey is not feasible under strict control measures during the epidemic. Instead, an online survey is the most appropriate data collection method for this particular period because it avoids the spread of the virus. Second, the cross-sectional design of this study prevents us from drawing causal links. Third, there may be inconsistencies between self-reported psychological status and professional clinical diagnoses due to the presence of recall bias and social desirability bias. Despite these limitations, our study is one of the few to investigate the psychological status of MSM population in Western China during COVID-19 and provides important information for the development of psychological interventions to mitigate the negative effects of COVID-19 on MSM population.

## 6. Conclusions

In summary, the sudden outbreak of the COVID-19 epidemic has had a negative impact on the mental health of the MSM population due to the changes in various aspects of their lives, which may have caused or even exacerbated a series of psychological problems. The incidence of psychological disorders in Chinese MSM is higher than that in other groups during COVID-19. Our study assessed the psychological status and influencing factors during COVID-19 epidemic in MSM population in Western China through PQEEPH. The results showed that factors associated with the COVID-19 may cause a range of mental health problems in this population. These findings can be used to target the development of psychological interventions to improve the mental health of MSM population during the epidemic and to avoid the severity and persistence of psychological problems, thereby improving their quality of life and social adjustment.

## Figures and Tables

**Table 1 ijerph-20-01333-t001:** Psychological Questionnaire for Emergent Event of Public Health (PQEEPH).

Dimensions	Items	Score
Depression	Cronbach’s α = 0.934	
(1)	Feeling dispirited, slowed down, inattentive, poor memory	1–4
(2)	Energy is worse than before	1–4
(3)	Mental fatigue and not easy to recover	1–4
(4)	No appetite and significant weight loss	1–4
(5)	The brain is not as flexible as before	1–4
(6)	Headache and muscle aches all over the body	1–4
Neurasthenia	Cronbach’s α = 0.905	
(1)	No interest in anything	1–4
(2)	Thinking without control	1–4
(3)	Feel annoyed and easily lose temper	1–4
(4)	Feeling that I am useless	1–4
(5)	Poor sleep (difficulty sleeping, dreamy, no relief after waking, sleep rhythm disorder)	1–4
Fear	Cronbach’s α = 0.774	
(1)	Fear of infection for yourself and your family	1–4
(2)	Repeatedly washing hands and scrubbing things, but always feel that they are not clean enough	1–4
(3)	I feel scared and my heart beats faster when I come across something related to COVID-19	1–4
(4)	Feeling nervous and anxious in places where people gather, especially near hospitals	1–4
(5)	Very concerned about any physical discomfort	1–4
(6)	Try not to go to hospitals or places where people gather, and always wear a mask when in contact with people	1–4
Anxiety	Cronbach’s α = 0.910	
(1)	Feeling a rapid heartbeat, sweating and blushing	1–4
(2)	Symptoms of dizziness, panic, bloating, constipation or diarrhea	1–4
(3)	I know it won’t help, but I can’t help thinking about it and washing my hands again and again	1–4
(4)	Unable to control excessive nervousness and fear	1–4
(5)	Want to die	1–4
(6)	Thinking about something related to COVID-19 doesn’t make you want to do anything else	1–4
Hypochondria	Cronbach’s α = 0.708	
(1)	Symptoms associated with COVID-19, suspected of being infected	1–4
(2)	Go to the hospital to determine if you are already infected	1–4
Total	Cronbach’s α = 0.961	

**Table 2 ijerph-20-01333-t002:** Demographic and HIV-related characteristics of participants (n = 412).

Variables	n	%
Demographic characteristics		
Age		
18~35	248	60.19
35~50	142	34.47
≥50	22	5.34
Household location		
Urban	388	94.17
Rural	24	5.83
Ethnicity		
Han ethnicity	373	90.53
Minority	39	9.47
Educational level		
High school or below	61	14.80
Vocational school	150	36.41
College or above	201	48.79
Employment status		
Employed	315	76.46
Unemployed/Retirement	73	17.72
Student	24	5.82
Marital status		
Married	68	16.50
Unmarried	344	83.50
Monthly personal income		
1000~5000 RMB	201	48.79
5000~10,000 RMB	139	33.74
≥10,000 RMB	72	17.47
HIV-related characteristics		
Sexual role		
Mainly “Top”	158	38.35
Both of it	72	17.48
Mainly “Bottom”	182	44.17
Number of male sexual partners in the last month		
0	227	55.10
1	123	29.85
≥2	62	15.05
Self-reported HIV infection status		
Positive	38	9.23
Negative	358	86.89
Unknown	16	3.88

HIV: human immunodeficiency virus.

**Table 3 ijerph-20-01333-t003:** Psychological status of the MSM population (n = 412).

Psychological Status Dimension	Score Range	Score (Mean ± SD)	Higher Risk of Mental Health Problem n (%)
Depression	6–24	9.91 ± 4.33	70 (16.99)
Neurasthenia	5–20	8.17 ± 3.50	58 (14.08)
Fear	6–24	11.27 ± 3.64	69 (16.75)
Anxiety	6–24	8.44 ± 3.64	68 (16.50)
Hypochondria	2–8	2.73 ± 1.14	75 (18.20)

MSM: men who have sex with men; SD: standard deviation.

**Table 4 ijerph-20-01333-t004:** Univariate analysis of participants’ COVID-19-related characteristics using five subscale high-risk scores of the PQEEPH (n = 412).

Variables	Higher Risk of Depression (n = 70)	Higher Risk of Neurasthenia (n = 58)	Higher Risk of Fear (n = 69)	Higher Risk of Anxiety (n = 68)	Higher Risk of Hypochondria (n = 75)
	*p*-Value (χ^2^)	*p*-Value (χ^2^)	*p*-Value (χ^2^)	*p*-Value (χ^2^)	*p*-Value (χ^2^)
COVID-19-related knowledge					
I am well-informed about the causes of COVID-19	**0.064** (3.42)	0.155 (2.02)	0.288 (1.12)	**0.082** (3.03)	0.487 (0.48)
I am well-informed about the transmission routes of COVID-19	0.388 (0.74)	0.499 (0.45)	0.254 (1.30)	0.142 (2.15)	**0.050** (3.80)
I am well-informed about the infectiousness of COVID-19	0.781 (0.08)	0.605 (0.27)	**0.081** (3.05)	0.644 (0.21)	**0.084** (2.99)
I am well-informed about the effectiveness of preventive measures for COVID-19	0.943 (<0.01)	0.246 (1.35)	0.947 (<0.01)	**0.084** (2.99)	**0.088** (2.91)
I am well-informed about the effectiveness of cure of COVID-19	0.327 (0.96)	**<0.001** (11.36)	**0.040** (4.20)	**0.047** (3.96)	**0.054** (3.70)
I am well-informed about the reinfection of COVID-19 after cure	0.359 (0.84)	**0.095** (2.79)	0.287 (1.12)	0.620 (0.25)	0.915 (0.01)
COVID-19-related attitudes					
I think COVID-19 is very contagious	0.567 (0.33)	0.984 (<0.01)	**0.014** (6.00)	0.893 (0.02)	0.379 (0.77)
I think COVID-19 is very prevalent where I live	0.213 (1.55)	0.110 (2.53)	0.215 (1.54)	**0.001** (10.63)	**0.017** (5.71)
I am afraid of COVID-19	**0.014** (6.00)	**0.029** (4.80)	**<0.001** (25.91)	**0.006** (7.51)	0.237 (1.40)
I think COVID-19 is very close to me	**0.013** (6.24)	**0.030** (4.67)	**<0.001** (20.16)	**0.002** (9.86)	**0.036** (4.42)
I felt nervous when I went out during the outbreak	**0.002** (9.46)	**0.016** (5.86)	**<0.001** (22.94)	**<0.001** (15.68)	**0.001** (10.18)
The outbreak is considered to be under effective control	**0.011** (6.51)	**0.027** (4.88)	0.336 (0.92)	**0.003** (8.70)	**0.002** (9.27)
I think there will be another small epidemic	**0.003** (8.66)	**0.019** (5.48)	**0.020** (5.43)	**0.029** (4.75)	**0.065** (3.39)
I think the outbreak will cause me financial losses	**0.040** (4.26)	**0.007** (7.24)	**0.007** (7.40)	0.269 (1.22)	**0.040** (4.22)
I think COVID-19 can be cured	0.674 (0.18)	**0.089** (2.90)	0.301 (1.06)	**0.098** (2.74)	**0.066** (3.39)
I think it is possible to be reinfected after COVID-19 is cured	**0.015** (5.88)	0.240 (1.38)	**0.051** (3.80)	0.142 (2.13)	**0.050** (3.84)
If I get infected, I will not be able to handle daily activities	**0.100** (2.69)	**0.034** (4.49)	**<0.001** (16.71)	**0.075** (3.17)	**0.068** (3.33)
I am very worried about another outbreak	**0.013** (6.18)	**<0.001** (12.53)	**<0.001** (25.13)	**0.033** (4.53)	**0.065** (3.40)
COVID-19-related behavior					
Had a COVID-19 test *	0.937 (0.01)	0.617 (0.25)	0.851 (0.04)	0.902 (0.02)	0.296 (1.09)
Risk perception					
Controllability					
I can take protective measures such as home isolation, wearing masks and washing hands regularly	**0.057** (3.63)	0.219 (1.51)	0.839 (0.04)	0.240 (1.39)	0.181 (1.80)
I can ensure that I am not infected with COVID-19	0.324 (0.97)	0.151 (2.06)	0.200 (1.64)	**0.037** (4.33)	**0.073** (3.20)
I can control the loss caused by COVID-19	0.465 (0.535)	**0.002** (9.551)	0.127 (2.341)	0.532 (0.384)	0.395 (0.723)
**Possibility**					
I’m very likely to get infected with COVID-19	0.685 (0.16)	0.544 (0.37)	0.696 (0.15)	0.906 (0.01)	0.725 (0.12)
**Susceptibility**					
I am more likely than others to get COVID-19	**0.015** (5.92)	**0.018** (5.59)	**0.003** (8.99)	**0.002** (9.49)	**0.005** (7.73)
**Severity**					
I think COVID-19 is very serious	0.380 (0.88)	0.429 (0.62)	**0.036** (4.44)	0.466 (0.52)	**0.030** (4.72)
**Epidemic exposure ***	**0.001** (10.74)	**0.007** (7.36)	**0.018** (5.64)	**0.024** (5.10)	**0.022** (5.23)

*: indicates that the Chi-Square Test was used; all other variables were tested with the Trend Chi-Square Test. Bold font represents *p*-values ≤ 0.10, which can be included in the multivariate logistic regression model. PQEEPH: Psychological Questionnaire for Emergent Event of Public Health.

**Table 5 ijerph-20-01333-t005:** Multivariate logistic regression analysis affecting the psychological status of MSM population during COVID-19 (n = 412).

Variables	cOR	95%CI	*p*-Value	AOR	95%CI	*p*-Value
Depression						
I am well-informed about the causes of COVID-19						
Not quite agree vs. Totally agree (reference)	2.08	0.88–4.89	0.041	2.69	1.06–6.80	0.020
Epidemic exposure						
Yes vs. No (reference)	2.45	1.42–4.24	0.001	2.26	1.27–4.02	0.006
Neurasthenia						
I am well-informed about the effectiveness of cure of COVID-19						
Totally disagree vs. Totally agree (reference)	7.81	2.57–23.74	<0.001	9.59	2.29–32.54	<0.001
**Fear**						
I am afraid of COVID-19						
Totally agree vs. Totally disagree (reference)	7.22	2.02–25.77	<0.001	4.71	1.27–17.56	<0.001
Anxiety						
I think COVID-19 is very prevalent where I live						
Totally agree vs. Totally disagree (reference)	3.00	1.29–6.99	0.001	2.79	1.12–6.92	0.003
I am more likely than others to get COVID-19						
Quite agree vs. Totally disagree (reference)	5.26	1.83–15.11	0.002	8.54	2.60–28.06	<0.001
I am well-informed about the causes of COVID-19						
Totally disagree vs. Totally agree (reference)	2.65	0.94–7.47	0.020	4.02	1.32–12.24	0.005
Hypochondria						
The outbreak is considered to be under effective control						
Totally disagree vs. Totally agree (reference)	7.09	1.51–33.24	0.054	28.47	2.59–312.67	0.017
I am more likely than others to get COVID-19						
Quite agree vs. Totally disagree (reference)	5.42	1.49–19.69	<0.001	4.81	1.70–13.58	<0.001

The regression model on Depression corrected for: age, marital status, and sexual role. The regression model on Neurasthenia corrected for: age, marital status, number of male sexual partners. The regression model on Fear corrected for: household location. The regression model on Anxiety corrected for: age, household location, sexual role. MSM: men who have sex with men; cOR: crude odds ratios; AOR: adjusted odds ratios.

## Data Availability

All data included in the current study can be accessed from the corresponding author through the email address of a rational request.

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
