# Peer review of "Psychological Status of Men Who Have Sex with Men during COVID-19: An Online Cross-Sectional Study in Western China"

_ijerph, 2023, doi:10.3390/ijerph20021333_

Round 1
Reviewer 1 Report
Dear auhors, congratulations for your hard work. It is true that the pandemic has affected us all, not only special groups of our fellow citizens. Such studies, like yours, contribute the most in understanding the kinetic and the impact of limitations in freedom, if I may say, such as a lockdown can force. However, it would be interesting to know how the participants have reacted to lockdown if they had a steady relationship and lived together, and even more if you had feedback of their partners also. Apart from a minor mistake in line 150 "Knowledge, attitudes, and behaviors related to COCID-19 were measured primarily..."where you have to replace COCID with COVID, I do not have anything else to comment.
Best of luck to your next project.
Reviewer 2 Report
Background, research data, and discussion are very well spelled out. I wish to make few suggestions:
Grammar and Spelling needs check at several places, e.g.,
Line 86. Can delete ", which"
Line 119. "MSMs were" instead of "MSMs are"
Lines 126-134. Tense may need to be corrected to past tense
Lines 150,279. COVID not COCID
The above are only examples of changes that need to be made.
Also,
Lines 320-330. Have these beliefs been tested in this study or is the affirmation of these beliefs within the scope of this study?. Line 363 seems to restrict this...
Conclusion can be lengthened a little.
Reviewer 3 Report
This is an interesting paper looking at the psychological status of Chinese men who have sex with men during the pandemic.
The paper concerns with an important and contemporary topic.
Although in my opinion this manuscript should be published, I would suggest some changes for improving it even more, as reported below. I hope that authors will find my suggestions in a constructive way.
INTRODUCTION
I would like to see a more expanded discourse on why and how exploring the psychological status of Chinese men who have sex with men under the COVID-19 is significant in terms of public policies.
THEORETICAL FRAMEWORK AND LITERATURE REVIEW
I feel that authors they should insert this part. They cannot fail to address some important issues, such as heteronormativity as a possible stressor for people.
I recommend placing their research within the theoretical framework of minority stress.
In addition, since the outbreak of the pandemic, a number of interesting papers bout consequences of COVID on people's lives have been published and it is important to mention them. Among the most important, I point out:
Choi, K. R., Heilemann, M. V., Fauer, A., & Mead, M. (2020). A second pandemic: Mental health spillover from the novel coronavirus (COVID-19). Journal of the American Psychiatric Nurses Association, 26(4), 340-343.
Gato, J., Barrientos, J., Tasker, F., Miscioscia, M., Cerqueira-Santos, E., Malmquist, A., ... & Wurm, M. (2021). Psychosocial effects of the COVID-19 pandemic and mental health among LGBTQ+ young adults: a cross-cultural comparison across six nations. Journal of Homosexuality, 68(4), 612-630.
Kathirvel, N. (2020). Post COVID-19 pandemic mental health challenges. Asian journal of psychiatry, 53, 102430.
Konnoth, C. (2020). Supporting LGBT communities in the COVID-19 pandemic. 2020). Assessing Legal Responses to COVID-19. Boston: Public Health Law Watch, U of Colorado Law Legal Studies Research Paper, (20-47).
Monaco, S. (2021). Tourism, Safety and COVID-19: Security, Digitization and Tourist Behaviour. Routledge.
METHODS
Authors should provide more information about the difficulties of doing research on a vulnerable population. On this aspect it might be useful to cite
Monaco, S. (2022). Gender and sexual minority research in the digital society. In Handbook of research on advanced research methodologies for a digital society (pp. 885-897). IGI Global.
DISCUSSION
The discussion is quite confusing, as it did not follow hypotheses. I suggest authors to rewrite the discussion on the basis of confirmation/non-confirmation of the hypotheses.
Similarly, suggested policy interventions seem disconnected from research results. The authors should strive to better connect the innovative elements of their work with possible policy actions by government and management.
Finally, I suggest to compare the results with respect to the suggest theoretical background and empirical research.
Round 2
Reviewer 3 Report
I am glad to see that the authors have given due consideration to my suggestions. The revised version of the paper is much improved and more readable. I suggest the authors read the text again carefully and correct typos and small errors, verifying that APA standards are met.
Please check in the bibliography the reference N. 34. It should be Monaco S. and not Salvatore M. (2021)